Feasibility of using microbeads with holographic barcodes to track DNA specimens in the clinical molecular laboratory

Merker Jason D. 1
O’Grady Naomi 2
Gojenola Linda 1
Dao Mai 1
Lenta Ross 2
Yeakley Joanne M. 2
Schrijver Iris 1 3 ischrijver@stanfordmed.org
1 Department of Pathology, Stanford University School of Medicine , Stanford, CA , USA
2 Illumina, Inc. , San Diego, CA , USA
3 Department of Pediatrics, Stanford University School of Medicine , Stanford, CA , USA
Arany Praveen
Electronic publication date: 2013 Jul 2
Publication date: 2013
Volume: 1
Electronic Location ID: e91
Received 2013 Apr 5; Accepted 2013 Jun 1
Copyright: © 2013 Merker et al.
Copyright year: 2013
Copyright holder: Merker et al.
License: This is an open access article distributed under the terms of the Creative Commons Attribution License, which permits unrestricted use, distribution, and reproduction in any medium, provided the original author and source are credited.
License URL: https://creativecommons.org/licenses/by/3.0/

Keywords: DNA extraction, Microbeads, Holographic barcode, Specimen identification, Molecular pathology, Quality, Quality assessment program

Funding: Stanford University Department of Pathology developmental research grant This work was supported by a Stanford University Department of Pathology developmental research grant. Veracode microbeads and consumables needed for the BeadXpress Reader System were provided by Illumina, Inc. The funders had no role in study design, data collection and analysis, decision to publish, or preparation of the manuscript.

==============================
We demonstrate the feasibility of using glass microbeads with a holographic barcode identifier to track DNA specimens in the molecular pathology laboratory. These beads can be added to peripheral blood specimens and are carried through automated DNA extraction protocols that use magnetic glass particles. We found that an adequate number of microbeads are consistently carried over during genomic DNA extraction to allow specimen identification, that the beads do not interfere with the performance of several different molecular assays, and that the beads and genomic DNA remain stable when stored together under regular storage conditions in the molecular pathology laboratory. The beads function as an internal, easily readable specimen barcode. This approach may be useful for identifying DNA specimens and reducing errors associated with molecular laboratory testing.

Introduction

Events that negatively impact, or could negatively impact, patient care because of quality concerns with DNA-based clinical genetic testing are rare, occurring in <0.5% of tests performed (Hofgartner & Tait, 1999). Despite this low rate of error, further reduction remains a priority of clinical laboratories. Errors can occur in all phases of testing (pre-analytical, analytical, and post-analytical), with the majority of problems occurring in the pre-analytical phase. This emphasizes the importance of quality assurance in all phases of testing. In this brief report, we describe the use of small glass microbeads containing a unique numeric code to barcode DNA eluates from peripheral blood specimens. The beads can be added directly to the peripheral blood specimens and are carried through automated DNA extraction protocols that use magnetic glass particles. A bead reader system can identify the bead number in the specimen, which can function as an internal barcode for specimen identification. This can be used to check the identity of DNA specimens that, for example, give unexpected results. Likewise, this approach could be used to check specimens at a set interval as part of the laboratory quality assessment program. Incorporation of this microbead system directly into blood draw tubes could be used to immediately barcode the specimen very early in the pre-analytical phase of testing. Preliminary data suggest that this approach can be used to track DNA specimens without interfering with DNA storage or downstream molecular testing. With the increased use of nucleic acid testing to guide precision or personalized medicine, this method provides a unique approach to decrease laboratory error.

Materials and Methods

Specimens

The use of peripheral blood specimens in this study was approved by a Stanford University IRB.

VeraCode glass microbeads

VeraCode microbeads were provided by Illumina, Inc. (San Diego, CA) in microcentrifuge tubes, each containing approximately 40,000 beads in 70% ethanol. VeraCode microbeads are cylindrical glass beads measuring 240 microns in length by 28 microns in diameter. A digital holographic element containing a numeric code is embedded within the beads serving as a unique identifier. When excited by a laser, each bead emits a unique code image that is detected by Illumina’s BeadXpress Reader System.

DNA extraction with VeraCode microbeads

Tubes containing the VeraCode microbeads were centrifuged in a microcentrifuge at greater than or equal to 10,000 rpm. Most of the 70% ethanol was removed from the tubes, leaving ∼150 µL of residual 70% ethanol with the 40,000 beads. Subsequently, 200 µL of peripheral blood was added to the 70% ethanol and bead mixture using a 1 mL pipet and mixed thoroughly. The peripheral blood specimens containing the bead mixtures were then extracted on the Qiagen BioRobot EZ1 Workstation using the EZ1 DNA Blood 350 µL Kit (Valencia, CA) following the manufacturer’s standard protocol with a 200 µL elution volume. The eluate containing the VeraCode beads was transferred to the well of a 96-well round bottom microplate (Corning Inc., Corning, NY). A KingFisher 96 pin magnet head with a tip comb (Thermo Fisher Scientific, Waltham, MA) was used to remove residual Qiagen magnetic beads that could potentially interfere with the BeadXpress Reader System.

Determination of the number of VeraCode microbeads carried through the DNA extraction process

On seven independent days, VeraCode microbeads were added to three peripheral blood specimens. As is outlined in Table 1, the same sets of peripheral blood specimens were used for two or three days. Following the extraction process described in the above section, the beads were transferred to a 76.2 × 25.4 mm standard glass microscope slide and a 24 × 50 mm cover glass was used. The VeraCode microbeads were counted using a microscope under 100 × magnification.

Table 1 Number of Veracode microbeads carried through the DNA extraction process.

Days 1, 2 and 3	Days 4 and 5	Days 6 and 7	
Specimen 1	Specimen 2	Specimen 3	Specimen 4	Specimen 5	Specimen 6	Specimen 7	Specimen 8	Specimen 9	
534	464	743	108	203	135	231	295	408	
1096	778	362	168	242	83	140	174	101	
105	279	257							

Utilization of VeraCode microbeads for specimen identification

Six peripheral blood specimens submitted for cystic fibrosis gene mutation testing underwent DNA extraction as described above with and without VeraCode microbeads added to the peripheral blood; beads containing a unique bead identification code were added to each peripheral blood specimen. DNA eluates with and without the beads were tested with the Cystic Fibrosis Genotyping Assay (Abbott Laboratories, Abbott Park, IL). In brief, this assay detects 32 cystic fibrosis transmembrane conductance regulator (CFTR) gene mutations via multiplex PCR amplification and oligonucleotide ligation assays followed by detection via capillary electrophoresis on an Applied Biosystems 3130xl Genetic Analyzer (Life Technologies Corporation, Carlsbad, CA). The resulting electropherograms were interpreted and evaluated for equivalency by two molecular pathologists (JDM and IS). A plate containing the residual DNA specimens with and without the beads was covered, frozen, and shipped to Illumina, Inc. to use the BeadXpress Reader System in order to determine the bead number associated with each specimen. We required that ≥ 20 beads with the same barcode be detected by the instrument for definitive specimen identification for all experiments. In internal experiments, Illumina has determined that the bead mis-identification rate is under 0.5%, suggesting that a requirement for 20 beads for sample identification is more than sufficient.

We selected six independent peripheral blood specimens from individuals without a known cancer diagnosis, and for three of the specimens, we spiked in a cell line containing the IGH-BCL2 translocation at a 1:10,000 dilution to generate positive sensitivity controls near the lower limit of detection of the IGH-BCL2 translocation assay. DNA extraction was performed as described above with and without VeraCode microbeads added to each of the six peripheral blood specimens. Beads containing a unique bead identification code were added to each peripheral blood specimen. DNA eluates with and without the beads were tested with our assay used to detect IGH-BCL2 translocations involving the major breakpoint region, minor cluster region and intermediate cluster region of BCL2. This assay uses multiplex, nested PCR followed by detection of amplification products via agarose gel electrophoresis. The resulting gel photographs were interpreted and evaluated for equivalency by two molecular pathologists (JDM and IS). A plate containing the residual DNA specimens with and without the beads was covered, frozen, shipped to Illumina, Inc., and scanned on the BeadXpress Reader System in order to determine the unique bead identification code associated with each specimen. The group using the BeadXpress Reader System was blinded to which specimens had beads added and to which bead identification code was used in each sample.

Examination of VeraCode bead and DNA stability

Three peripheral blood specimens were used for this portion of the study – each was submitted to our molecular pathology laboratory for testing with either the Cystic Fibrosis Genotyping Assay described above, the quantitative JAK2 V617F MutaQuant assay (Ipsogen, Stamford, CT), or our laboratory-developed Fragile X syndrome assay. The JAK2 V617F MutaQuant assay is an allele-specific, real-time PCR assay that quantifies both JAK2 V617F and corresponding wild-type alleles. Our laboratory-developed Fragile X syndrome assay uses PCR optimized for GC-rich amplicons followed by capillary electrophoresis on an Applied Biosystems 3130xl Genetic Analyzer to detect (CGG)n trinucleotide repeat expansions in the 5′ untranslated region of FMR1. Each specimen was assigned a unique bead identification code. Beads with this identification code were added to four aliquots of each peripheral blood specimen, and the specimens were extracted as described above. Eluates containing the beads were incubated at 25°C. Two aliquots of each specimen were removed after 50 days and 90 days. Each specimen was re-tested using the same assay for which the specimen was originally submitted, and the results were interpreted and evaluated for equivalency by two molecular pathologists (JDM and IS). A plate containing the DNA specimens with beads was covered, frozen, and shipped to Illumina, Inc. to use the BeadXpress Reader System in order to determine the bead identification code associated with each specimen. The group using the BeadXpress Reader System was blinded to which bead identification code was added to each specimen.

Accelerated stability calculations used the Q10 model, a simplification of the Arrhenius equation approach (Hemmerich, 1998). In this experiment, the method predicts the stability of a well-characterized reagent at ≤ 4°C based on stability at the elevated temperature of 25°C. The Q10 value equals 2. Under the Q10 model, the time relationship between the accelerated temperature (Taccel) and the storage temperature (Tstorage) is given by the equation Q Factor = 2((Taccel−Tstorage)/10); therefore, for this experiment the Q Factor = 2((25−4)/10) = 4.3. One day at 25°C is equivalent to ∼4 days at 4°C.

Results

We initially observed that VeraCode microbeads containing a holographic identification code could be added to whole blood specimens and subsequently carried through the genomic DNA extraction process on instruments that use magnetic glass particles. We observed equivalent results using the BioRobot EZ1 Workstation (Qiagen) and MagNA Pure LC (Roche Diagnostics Corporation, Indianapolis, IN), and used the BioRobot EZ1 Workstation for subsequent work described in this manuscript. Given potential applications for specimen identification and quality assurance in the molecular pathology laboratory, we performed a set of experiments designed to evaluate the potential feasibility of using the VeraCode microbeads to track specimens in a clinical molecular laboratory. We found that a sufficient number of microbeads are consistently carried over during genomic DNA extraction to allow specimen identification, that the beads do not interfere with several different molecular assays, and that the beads and genomic DNA are stable when stored together over extended periods of time.

We first evaluated if a sufficient number of microbeads would be carried through the DNA extraction procedure to be read by the BeadXpress Reader System (Illumina), thereby allowing identification of the specimen being tested. Although in practice even a single bead can be read and detected by the system, we conservatively established that at least 20 microbeads with the same identification code should be present to allow reliable detection with this system. As is shown in Table 1, well in excess of 20 microbeads were carried through the extraction procedure using multiple different specimens over multiple days. In total, 21 individual extractions were evaluated, yielding a median of 242 microbeads per extraction (range 83–1096). Given that 40,000 microbeads are added to the initial whole blood aliquot, the efficiency of carryover to the DNA eluate is low, but a sufficient number of beads are consistently transferred to allow detection by the system. Of note, the measured DNA yield of extractions that included the beads was generally lower than that of the same specimen extracted without microbeads – average of 57 ng/µL without beads and 34 ng/µL with beads (Table 2). PCR can be reproducibly performed on nanogram quantities of genomic DNA, and our laboratory generally uses between 1 and 1,000 ng of genomic DNA per PCR reaction. Therefore, the DNA concentration and total yield were for sufficient for molecular testing, and we usually had micrograms of residual DNA. It is unclear why the presence of the microbeads during the extraction process reduced the amount of DNA in the eluate. It is possible that some of the DNA was bound to the glass microbeads and therefore not available to be measured in solution. Collectively, these data indicate that a sufficient number of microbeads are carried through the DNA extraction procedure to be read by the BeadXpress Reader System, and that inclusion of the microbeads does not appreciably impact the DNA extraction process.

Table 2 Effect of including Veracode microbeads during DNA extraction on resulting DNA concentration.

Specimen	DNA concentration when
extracted without beads	DNA concentration when
extracted with beads	
1	43 ng/µL	22 ng/µL	
2	51 ng/µL	33 ng/µL	
3	44 ng/µL	24 ng/µL	
4	73 ng/µL	44 ng/µL	
5	49 ng/µL	25 ng/µL	
6	79 ng/µL	53 ng/µL	

We subsequently examined whether inclusion of the VeraCode microbeads affected the performance of two clinical assays commonly performed in our molecular pathology laboratory and whether the beads could be used to track DNA specimens within our laboratory. We hypothesized that the beads would not interfere with molecular assays because they are commonly used in related genotyping assays (Lin et al., 2009). Six peripheral blood specimens submitted for CFTR mutation testing underwent DNA extraction with and without VeraCode microbeads added to the peripheral blood. A unique bead identification code was added to each peripheral blood specimen, essentially adding a readily readable molecular barcode to the specimen. DNA eluates with and without the beads were tested with our laboratory’s cystic fibrosis carrier screening assay. The resulting data were interpreted by two molecular pathologists, and the results of the cystic fibrosis assays were the same in the paired peripheral blood specimens with and without the beads (Table 3). In addition, peak positions and heights were equivalent in the specimens with and without beads (Fig. 1 illustrates a representative example), indicating that the presence of the microbeads does not significantly interfere with the performance of this assay. Subsequently, members of our group blinded to which bead was added to each specimen successfully used the BeadXpress Reader System to identify the correct bead identification code originally added to each specimen (Table 3).

Table 3 Using Veracode microbeads to identify DNA specimens and examining their effect on two molecular pathology assays.

Specimen
number	Bead number
added	Bead number
detected	Assay	Assay result with microbeads
added	Assay result without beads
added	Equivalent
amplification	
10	4096	4096	CF32	homozygous delF508	homozygous delF508	Yes	
11	272	272	CF32	homozygous delF508	homozygous delF508	Yes	
12	40	40	CF32	homozygous delF508	homozygous delF508	Yes	
13	136	136	CF32	no mutations detected	no mutations detected	Yes	
14	2560	2560	CF32	no mutations detected	no mutations detected	Yes	
15	1040	1040	CF32	no mutations detected	no mutations detected	Yes	
16	1040	1040	BCL2	IGH-BCL2 detected	IGH-BCL2 detected	Yes	
17	2560	2560	BCL2	IGH-BCL2 not detected	IGH-BCL2 not detected	Yes	
18	136	136	BCL2	IGH-BCL2 detected	IGH-BCL2 detected	Yes	
19	40	40	BCL2	IGH-BCL2 not detected	IGH-BCL2 not detected	Yes	
20	272	272	BCL2	IGH-BCL2 detected	IGH-BCL2 detected	Yes	
21	4096	4096	BCL2	IGH-BCL2 not detected	IGH-BCL2 not detected	Yes	
Notes.

CF32 – Abbott Laboratories Cystic Fibrosis Genotyping Assay; BCL2 – Laboratory-developed assay for detection of IGH-BCL2 translocations.

Figure 1 Representative capillary electropherograms from a multiplex PCR amplification and oligonucleotide ligation assay to detect 32 different mutations in the CFTR gene.

The presence of beads during the extraction process and downstream steps (A) does not appear to affect either peak height or assay results when compared to analysis of the same specimen without beads (B). No CFTR mutations were detected in the specimen with or without the beads and each of the peaks is present at the same position.

We also performed a similar set of experiments to that described above with the cystic fibrosis carrier assay, using our laboratory assay for the qualitative detection of IGH-BCL2 translocations by PCR. As is seen in Table 3 and Fig. 2, the results were the same for all six specimens tested and assay performance was equivalent. Likewise, we were able to re-identify in a blinded manner which bead was added to which specimen.

Figure 2 Agarose gel electrophoresis demonstrating control amplification reactions for a PCR-based IGH-BCL2 translocation assay.

The presence of beads during the extraction process and downstream steps does not significantly affect the control amplification for this assay. This control amplicon is a 270 bp product derived from the F5 gene.

The cystic fibrosis and IGH-BCL2 experiments also allowed examination of time required for operation of the BeadXpress Reader System. Since the plates containing the specimens with microbeads can be directly loaded into a drawer in the BeadXpress Reader System after thawing, minimal hands-on time is required for this step. Once the instrument is initialized, the time to get results for 1, 12, or 96 specimens is approximately 5, 10, and 60 min, respectively.

Finally, to examine the long-term stability of the VeraCode microbeads we used accelerated stability experiments, and we also evaluated the integrity of the DNA under these conditions. We selected three peripheral blood specimens submitted to our molecular pathology laboratory for testing with either the Cystic Fibrosis Genotyping Assay, a quantitative JAK2 V617F MutaQuant assay, or our laboratory-developed Fragile X syndrome assay. Beads with a number specific to each specimen were added to multiple aliquots of each peripheral blood specimen, and DNA was extracted from the specimens. Eluates containing the beads were incubated at 25°C, which was our elevated storage temperature. Using the Q10 model, one day at 25°C is equivalent to just over four days at or below 4°C. Two aliquots of each specimen were removed from incubation at 25°C after 50 days and 90 days, which is projected to represent 4°C incubation for 215 days and 387 days. Each specimen was re-tested using the same assay for which the specimen was originally submitted, and the results were equivalent in both replicates at both time points. In addition, the correct bead number associated with each replicate was identified in a blinded manner. We note that the accelerated stability calculations using the Q10 model is a conservative approach in the assignment of Q10 = 2. In addition, if this method is used to calculate the stability of reagents stored at −15°C or below, additional stability may be conferred due to the phase transition. Collectively, these data indicate that the beads and the DNA in the eluate are stable for at least one year at 4°C and possibly longer when stored frozen.

Discussion

In this report, we evaluated the feasibility of using the VeraCode microbeads to track DNA specimens in the clinical molecular laboratory. We found that a sufficient number of microbeads are consistently carried over during genomic DNA extraction to allow specimen identification, that the beads do not appear to interfere with several different molecular assays, and that the beads and genomic DNA are stable when stored together over extended periods of time. The beads function as an internal, easily readable specimen barcode. This method may provide an additional approach to identifying DNA specimens and minimizing errors associated with molecular laboratory testing. Presently, this approach could be used to recheck the identity of DNA specimens that give unexpected results or to check specimens at a set interval as part of the laboratory quality assurance program. This would be a novel way to monitor for possible specimen mix-ups. Although the hands-on time and scanning time are reasonable, our experience suggests that it would not be practical to examine every specimen tested by a busy molecular pathology laboratory with our method.

We suggest that this approach represents a novel mechanism to track DNA specimens in the molecular pathology laboratory, and the data presented provide a proof of concept that such an approach is possible. However, a couple of technical issues related to the microbead size currently limit the potential utility of this approach. The present microbeads are too large to co-elute with DNA using column-based extraction methods, and consequently this microbead tracking system cannot be used with many DNA and RNA extraction methods. Likewise, the relatively large size of the microbeads makes it difficult to consistently transfer a sufficient number of beads from the DNA eluate to downstream assay steps due to the microliter volumes typically used during molecular testing. As an example, we found that addition of DNA from the tube containing the DNA eluate with microbeads to a standard PCR reaction did not reliably result in the transfer of a sufficient number of beads to be detected by the barcode reader. However, further manipulation of the beads or extraction process may allow for these limitations to be overcome and for this application to become more broadly applied.

Other factors must be addressed and studies performed before the method can be incorporated into routine clinical practice. First, the BeadXpress Reader System and associated VeraCode microbeads were not designed for this application. Consequently, the final retail list price of the reader, $98,000, was likely higher than an instrument designed solely for applications described in this manuscript. Furthermore, the configuration of VeraCode microbeads used in this study was custom prepared and is not commercially available at this time. Second, we evaluated this method to track DNA specimens extracted from peripheral blood. For this method to be applied to all areas of the molecular pathology laboratory, further examination of RNA extracted specimens would need to be performed. Likewise, specimens other than peripheral blood would need to be evaluated, and in the case of tissue specimens, the protocol would likely need to be modified to avoid, among other issues, destruction of the glass microbeads during the homogenization process. Third, we examined four assays in this study and a thorough validation would require examination of a larger number of qualitative and quantitative assays with associated statistical assessment of interference.

In summary, we have demonstrated a proof of concept that glass microbeads with a holographic numeric code can be used to barcode DNA eluates from peripheral blood specimens. This represents a novel way to track DNA specimens in the molecular pathology laboratory, and we suggest that with appropriate modifications and further evaluation such an approach could minimize the potential for errors associated with molecular laboratory testing.

Additional Information and Declarations

Competing Interests

Author Contributions

Human Ethics

Naomi O’Grady, Ross Lenta, and Joanne M. Yeakley were employees of Illumina, Inc. during this study. Iris Schrijver is an Academic Editor for PeerJ.

Jason D. Merker conceived and designed the experiments, performed the experiments, analyzed the data, contributed reagents/materials/analysis tools, wrote the paper.

Naomi O’Grady and Joanne M. Yeakley conceived and designed the experiments, contributed reagents/materials/analysis tools, wrote the paper.

Linda Gojenola conceived and designed the experiments, performed the experiments.

Mai Dao performed the experiments.

Ross Lenta conceived and designed the experiments, performed the experiments, contributed reagents/materials/analysis tools.

Iris Schrijver conceived and designed the experiments, analyzed the data, wrote the paper.

The following information was supplied relating to ethical approvals (i.e., approving body and any reference numbers):

The use of peripheral blood specimens in this study was approved by a Stanford University IRB (protocol 8353).

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
