# Peer review of "Feasibility of using microbeads with holographic barcodes to track DNA specimens in the clinical molecular laboratory"

_PeerJ, doi:10.7717/peerj.91_

## Round 0.1 · original submission · Minor Revisions

Please address the reviewer 2's concern about amount of DNA absorbed to the beads.

Reviewer 1 ·

Basic reporting

This is a well written proof of concept article on a novel way to track specimens through a molecular lab.

Experimental design

The study is well designed and the methods are sufficient to address the study goals.

Validity of the findings

In this proof of concept these authors were able to demonstrate the feasibility of using microbeads to track laboratory specimens. They appropriately address current limitations of this technique and make valid suggestions for how to further validate this process to make it more widely applicable to different sample types and tests.

·

Basic reporting

The article by Merker et. al, titled :"Feasibility of using microbeads with holographic barcodes to track DNA specimens in the clinical molecular laboratory", presents report of the proof-of principle studies for the identification of DNA samples with holographic glass beads. The authors show that these samples are stable over significantly long periods of time (days and weeks) and that DNA extracts produce specific readouts both in the gene-specific assay, while retaining sufficient amounts of beads for the sample identifications. Article is written clearly, figures are well made and related to the data presented in the main text.
Discussion is sufficient to the scope of the article, and presents a perspective on practical applications of the method described.

Experimental design

Experimental design is clear, and presents several examples of the gene identification assays, based on two methods of DNA extraction. Analysis of the DNA samples is thorough, having sufficient controls in each study.
I would like to suggest the minor control experiment, that may determine the non-specific absorption of the extracted DNA to the beads, which may be the limiting step in the overall efficiency of the process.

Validity of the findings

The findings for the proof-of-principle studies are sufficient and valid. Discussion had addressed future directions and possible limitations of the presented method.

---

## Round 0.2 · accepted · Accept

Thank you for addressing the reviewers comments appropriately. We look forward to your future involvement with the journal.